# Excessive Knee Internal Rotation during Grand Plié in Classical Ballet Female Dancers

**DOI:** 10.3390/sports12020054

**Published:** 2024-02-07

**Authors:** Aspasia Fotaki, Athanasios Triantafyllou, Panagiotis Koulouvaris, Apostolos Z. Skouras, Dimitrios Stasinopoulos, Panagiotis Gkrilias, Maria Kyriakidou, Sophia Stasi, Dimitrios Antonakis-Karamintzas, Charilaos Tsolakis, Olga Savvidou, Georgios Papagiannis

**Affiliations:** 11st Department of Orthopaedic Surgery, National and Kapodistrian University of Athens, 12462 Athens, Greece; athanat@gmail.com (A.T.); info@drkoulouvaris.gr (P.K.); apostolis.sk@gmail.com (A.Z.S.); jkaramintzas71@gmail.com (D.A.-K.); tsolakis@phed.uoa.gr (C.T.); olgasavvid@med.uoa.gr (O.S.); grpapagiannis@yahoo.gr (G.P.); 2Biomechanics Laboratory, Department of Physiotherapy, University of the Peloponnese, 23100 Sparta, Greece; p.gkrilias@uop.gr (P.G.); m.kyriakidou@uop.gr (M.K.); 3Laboratory of Neuromuscular and Cardiovascular Study of Motion, Physiotherapy Department, Faculty of Health and Care Science, University of West Attica, 12243 Egaleo, Greece; dstasinopoulos@uniwa.gr (D.S.); soniastasi1@gmail.com (S.S.); 4Sports Performance Laboratory, School of Physical Education & Sports Science, National and Kapodistrian University of Athens, 17237 Athens, Greece

**Keywords:** dance biomechanics, classic ballet, Grand Plié, 3D motion analysis

## Abstract

Classical ballet dancers are exposed daily to physically demanding movements. Among these, the Grand Plié stands out for its biomechanical complexity, particularly the stress applied to the knee joint. This study investigates the knee kinematics of healthy professional classical ballet dancers performing the Grand Plié. Twenty dancers were evaluated with a motion analysis system using a marker-based protocol. Before measurements, the self-reported Global Knee Functional Assessment Scale was delivered for the knees’ functional ability, and the passive range of knee motion was also assessed. The average score on the Global Knee Functional Assessment Scale was 94.65 ± 5.92. During a complete circle of the Grand Plié movement, executed from the upright position, the average maximum internal rotation of the knee joint was 30.28° ± 6.16°, with a simultaneous knee flexion of 134.98° ± 4.62°. This internal rotation observed during knee flexion exceeds the typical range of motion for the joint, suggesting a potential risk for knee injuries, such as meniscal tears. The findings provide an opportunity for future kinematic analysis research, focusing on the movement of the Grand Plié and other common ballet maneuvers. These data have the potential to yield valuable information about the knee kinematics concerning meniscus damage.

## 1. Introduction

The knee joint, a very intricate structure, is a well-studied area of interest due to its functional complexity and prevalence as the second most frequently injured region among athletes [1]. Various soft tissues and structures, including the lateral and medial menisci, play roles in the correct biomechanics and long-term anatomical well-being of the knee joint [2]. In a study conducted by Majewski et al. [3], a total of 17,397 individuals were documented as having experienced 19,530 sports-related injuries throughout a span of ten years. Medial and lateral meniscus tears accounted for 14.5% of these sports-related injuries. Common activities leading to meniscus tears include soccer, skiing, judo, handball, volleyball, basketball, American football, gymnastics, tennis, trekking, and dance [3,4].

Ballet, as a dance form, involves complex and physically demanding movements, including jumps, turns, and pointe work, which require significant strength, flexibility, and control. These biomechanical demands are comparable to those in many recognized sports [5,6]. A study on elite adolescent ballet dancers found that their dance volume ranged from 19.26 to 22.17 h per week across three training years [7]. In The Royal Ballet (London, UK), differences in weekly dance hours were observed between company ranks, ranging from 19.1 to 27.5 h per week [8]. Professional ballet dancers can range in age from adolescence to mid-30s or older [9,10]. While there is ongoing debate regarding the classification of ballet as a sport or an art, current evidence increasingly supports its recognition as a sport due to its rigorous physical demands and training requirements, akin to those found in competitive athletic disciplines [6,11]. The World DanceSport Federation (WDSF), which is recognized by the International Olympic Committee (IOC), oversees competitive dance sports. Although ballet is not directly governed by WDSF, the athletic aspects of ballet are widely recognized in the dance and sports communities, such as the World Ballet Competition and the United States Dance and Sport Competition Federation.

Several studies have shown a correlation between ballet dancing and a higher incidence of lower limb injuries [12,13,14,15,16], particularly at the knee joint [17]. In a study of 3482 cases of internal knee trauma, only 32% were attributed to meniscus injuries. However, when comparing the total number of knee traumas among active members of regional sports clubs, the incidence was similar between dancers, soccer players, and volleyball players. Notably, dancers exhibited a higher proportion of meniscus injuries, with 15 out of 27 total injuries (55%) involving the meniscus (6 lateral and 9 medial), compared to 31% in soccer and 34% in volleyball players [3]. Other epidemiological studies have shown that the incidence of knee joint/ligament derangement in female athletes is 0.19 [95%CI: 0.10–0.39] per 1000 training hours, and nearly 7% of all injuries occur during Plié/Relevé movements [18]. Also, the majority of injuries in ballet dancers happen during rehearsals [18]. This association can be attributed to the rigorous nature of ballet training, which involves intense physical exertion, repetitive “squatting” movements, demanding leaps, and the execution of specific musculoskeletal techniques. The dancer is prone to injuries in the hip, knee, ankle, and foot regions as they constantly strive to perfect their technique over time [13]. Risk factors involve several elements, such as an improper technique, excessive or insufficient training, sickness, and an inadequate training environment [14]. The knee joint is susceptible to meniscus tears or detachments when performing choreography or repetitive motions [19]. From imaging studies, it has been revealed that meniscal tears are among the most frequent injuries in elite dancers [20]. According to a case study conducted by Marios et al., it was observed that meniscus tears can occur while dancing, even in cases when other structures remain intact [21].

During training activities, various contact or non-contact mechanisms are related to meniscus tears, with the latter being the most frequent. Non-contact mechanisms leading to meniscus tears often arise from a particular kinematic pattern that involves a flexed knee joint and the application of excessive rotational forces [3,22].

Ballet, a sport known for its physical demands, entails the execution of specific repetitive motions such as the Grand Plié. The Grand Plié is a fundamental movement in ballet, often utilized as a transition between other movements. However, it is also a critical stand-alone exercise that is integral to ballet training and performance. The Grand Plié, therefore, serves a dual role in ballet, both as a transition and as a fundamental technique. In classical ballet, first, the dancer executes the Grand Plié motion from an upright stance, commonly referred to as the first position, with the feet firmly grounded. The hips are externally rotated, the knees hyperextended, and the pelvis is posteriorly tilted. The upper body is perfectly aligned so that the center of gravity is located between the two heels. Next, keeping this position firmly, the dancer begins to bend their knees and laterally rotate their hips even further. Finally, upon reaching the apex of the motion, the athlete reverts to the initial position while maintaining stability and control in all body segments [23,24,25].

Grand Plié biomechanics replicate the mechanism of a meniscus injury [23], suggesting a potential explanation for the increased rate of such injuries in ballet dancers [24,26,27]. The knee joint plays a crucial role in facilitating an axial rotational mechanism that enables the execution of flexion and extension movements with a complete range of motion. Typical knee motion requires the coupled rotation of either the femur or tibia along a flexion and extension motion, starting from the initial 5 degrees of flexion, due to the anatomical structure of the medial condyle. As the degree of passive flexion increases within the 30–90 degrees range, there are observed external and internal passive rotations of approximately 45° and 25°, respectively [28]. A similar pattern has been observed during simulated active knee flexion during squats, with the tibial rotation being higher at 90–120 degrees compared to the initial degrees of knee flexion [29].

Previous research has indicated that the specific biomechanics of the Grand Plié result in high ground reaction forces and excess joint movement in the sagittal plane [30,31,32]. Although pelvis [33], hip [34], and knee [35] kinematics have been assessed in the Grand Plié, no study seems to have measured knee kinematics in the transverse plane concerning the maximum knee flexion during the Grand Plié. This study aims to investigate knee rotation throughout flexion in the Grand Plié movement in professional classical ballet dancers. We anticipate that our findings will align with the range observed in previous studies on similar movements, such as squats, thereby contributing to a broader understanding of knee mechanics in dance movements.

## 2. Materials and Methods

### 2.1. Subjects

Twenty healthy professional female classical ballet dancers with a mean age of 23.6 years participated in the study (Table 1). All participants were members of the Greek National Opera Ballet and had a minimum of 8 years of training history, averaging 14.05 ± 4.86 years. The daily training volume was identical for all the dancers, involving five days a week of training, with each session lasting from approximately three to four hours. The exact age in years was calculated by subtracting each participant’s date of birth from the date of measurement, and then dividing the result by 365:Age = Date of measurement − Date of birth)/365

The inclusion criteria for the study were as follows: professional female classical ballet dancers in good health and no history of knee joint injuries, overuse disorders, or surgeries on the lower extremities in the past year. The exclusion criteria included recent injuries or conditions that could potentially affect the study outcomes. Accordingly, two subjects were excluded from the final analysis due to the presence of a first-degree ankle sprain that occurred during their daily training and prior to the commencement of the measurements. However, the study did include one participant who had a posterior horn medical meniscus tear two years prior to the assessment and another who had experienced a second-degree ankle sprain about five years before the measurements. The study was approved by the ethical committee of a university hospital under the approval number (EBΔ538/4-10-2021), and every participant provided their informed consent by signing the appropriate forms.

### 2.2. Knee Symptoms and Functional Assessment

To start, the participants completed a self-assessment scale to evaluate the functional abilities of their knees using the Global Knee Functional Assessment Scale. The scale ranged from 0 to 100, where a score of 100 denoted the highest level of functional capacity of the joint, absent of any indications of pain, instability, or restricted range of motion (ROM).

### 2.3. Clinical Examination—Passive Knee ROM

The knee’s passive ROM was measured using a standard long-arm goniometer (Gollehon Extendable Goniometer—Model 01135, Lafayette Instrument, Lafayette, IN, USA). Passive ROM was assessed by one clinician with more than ten years of experience. Regarding the measurement reliability, our study utilized long-arm goniometry, which is proven to have high inter- and intra-rater reliability, with an Intraclass Correlation Coefficient (ICC) greater than 0.99 for both [36,37]. The goniometer was placed at the joint center, determined by the clinician’s palpation on recommended bony landmarks, lateral femur epicondyle, along the femur to the greater trochanter (stationary arm), and along the fibula to the lateral malleolus (movement arm) [38]. Knee extension with a parallel tibia and femur was considered as 0 degrees. The measurement of passive knee extension was conducted with the foot supported on the treatment surface while the dancer assumed a supine position. Positive values were used to define the flexion position of a fully extended knee, while negative values were used to define hyperextension. While supine, the clinician moved the patient’s heel toward the pelvis to determine the knee’s maximum flexion. Internal and external rotation passive ROMs were measured with the subject seated and the knee flexed to 90 degrees, using the same goniometer. The clinician rotated the tibia internally, then externally, and recorded the corresponding passive ROM values. The neutral position of 0 degrees rotation was established as the resting position of the knee for each participant, with the tibia being vertical to the floor.

### 2.4. Instrumentation and Procedure

All measurements were performed at a motion and gait analysis laboratory placed in a university hospital facility. The room temperature was 20–25 °C, regulated to facilitate the warm-up of the dancers. The warm-up consisted a standardized routing of typical ballet exercises, for a time of five to ten minutes. Various equipment, including floor mats and elastic exercise bands, were readily accessible for the purpose of warming up the volunteers. The intensity of the light provided was enough, and its frequency was within a range that did not have any adverse impact on the process of data collecting. A well-lit level floor free of obstruction was used.

A 3D motion capture system was used with 8 high-resolution optoelectronic cameras (SMART-DX 4000, BTS Bioengineering, Garbagnate Milanese, Italy), with a sensor resolution of 2048 × 2048 pixels, accuracy of < 0.1 mm on a volume of 6 m × 6 m × 3 m, fixed focal length lens of 4.5–8 mm, and zoom lens of 6–12/25 mm. The superior accuracy of the optoelectronic measurement systems in capturing human sports movement is highlighted in the comprehensive review by van der Kruk and Reijne [39].

The equipment calibration was performed each morning by the same biomechanist in accordance with the established local standards. Test documentation and corresponding calibrations were carefully retained alongside the session data [40].

Following the completion of the warm-up phase, a total of twenty-two markers were carefully placed over the dancer’s body in accordance with the Davis protocol [41]. To perform the Davis protocol appropriately, a caliper was utilized to measure the diameters of the knee joint and the ankle, anterior superior iliac spine distance, and pelvic depth. The dancers were instructed to enter the first position of classical ballet to execute the Grand Plié with their feet positioned in the center of a force plate. The Grand Plié was executed without the barre to isolate and highlight the relevant knee movements without external support. The “Adagio center” music facilitated uniform rhythm and tempo during the movement execution for all participants. The dancers had to maintain their balance while securely placing their feet in the first position. Each trial round was considered to be complete when the dancer successfully transitioned from a full flexion Grand Plié to the finishing upright position (Figure 1). Periods of rest were provided between trials. To prevent the inclusion of unrepresentative data trials, all acquisitions were carefully reviewed for consistency in Grand Plié pace, as well as any inappropriate technical deviations resulting from fatigue or distraction. The initial full trial was assessed to verify the accurate reconstruction of all markers. A predetermined minimum of five Grand Plié cycles were obtained and assessed for marker continuity. The kinematic curves were examined to ensure their representativeness and consistency during the athlete’s presence. Three Grand Plié cycles were ultimately selected for further analysis. The same biomechanist conducted the entire process.

Basic statistical computations were performed using a spreadsheet software (Excel 365, Microsoft Co., Redmond, WA, USA). Sagittal and transverse plane knee kinematics were analyzed. Descriptive statistics are presented as mean and standard deviation (SD).

## 3. Results

### 3.1. Knee Symptoms and Functional Assessment

The average score of the Global Knee Functional Assessment Scale was 94.6 (SD: ±5.9), indicating a high functional level of the knee.

### 3.2. Clinical Examination of ROM

The passive knee flexion was measured at 156.08° (SD: ±3.02°) and the passive knee extension at −1.7° (SD: ±0.53°). The passive internal and external rotations were 25.04° (SD: ±2.28°) and 41.29° (SD: ±1.96°), respectively (Figure 2 and Figure 3).

### 3.3. Grand Plié Kinematic Measurements

During a complete circle of Grand Plié movement, the dancers demonstrated an average maximum internal rotation of the knee joint at 30.28° (SD: ±6.16°), with a simultaneously observed mean knee flexion at 134.98° (SD: ±4.62°) (Figure 4).

## 4. Discussion

The Grand Plié is one of the fundamental exercises for ballet dancers, and its biomechanical characteristics are of particular research interest. Previous research has been focused mainly on pelvic and hip kinematics, while the knee has been studied only in the sagittal plane. Indeed, when performing the Grand Plié exercise, the dancers exhibited a maximum mean knee internal rotation of 30.28 degrees at a knee flexion angle of 134.98 degrees, as measured in our research. Neumann [42] states that the typical range for maximum total knee joint rotation is between 40 and 45 degrees, which normally occurs when the knee joint is flexed at an angle of 90 degrees. The external to internal rotation ratio is 2:1, with the variables set at 27–30 degrees and 13–15 degrees, respectively [42]. Our research suggests that the Grand Plié movement might be linked to a higher risk of knee injuries due to its extensive internal rotation. Our research results assume that the Grand Plié movement can contribute to the risk of knee joint injuries due to the high value of knee internal rotation ROM.

According to previous research, axial rotation is an integral component of full knee flexion and extension motion, as demonstrated in the study by Zarins, Rowe, Harris, and Watkins [28]. They passively measured seven subjects with full knee ROM and functional ability, uncovering that, for 30–90 degrees of flexion, the knee exhibited approximately 45 degrees of normal external rotation and 25 degrees of normal internal rotation. This rotational motion diminished as the knee extended, showing 23 degrees of external rotation and 10 degrees of internal rotation at a flexion angle of 5 degrees [28]. Our study aligns with these findings, revealing that passive rotations for professional ballet dancers are within these limits, with 25.04° ± 2.28° for internal rotation and 41.29° ± 1.96° for external rotation. Diverging from previous methods, our research utilized a weight-bearing activity to assess rotation, more accurately reflecting the actual physical demands an athlete experiences. Data were collected from participants in an upright to fully flexed position during the Grand Plié, offering a robust framework for conducting a 3D kinematic study. This approach not only reflects real-world training conditions, but also indicates an excessive internal rotation of the knee when the flexion angle exceeds 120 degrees.

Foot position affects the rotational movements of the knee [43]. To evaluate the impact of excessive foot pronation on the knee’s transverse range of rotational motion, Coplan [44] used the Cybex II isokinetic dynamometer. The study included a sample of 15 individuals with normal foot pronation and 15 individuals with pronating feet. Tibial rotation was measured at three distinct knee flexion angles: 5, 15, and 90 degrees. A positive correlation was found between the degree of knee flexion and the degree of tibial rotation. At an angle of 90 degrees, the normal group had a passive mean range of tibial rotation of 32.1 degrees, whereas the pronating group demonstrated a range of 35.7 degrees. An excessive range of motion in tibial rotation, in conjunction with laxity in joint structures at broader angles of the knee joint, has been identified as a potential risk factor for knee joint injury [44]. This clinical observation is further supported by a comprehensive analysis that establishes a correlation between the joint laxity of ballet dancers and many intrinsic variables, resulting in impaired proprioception and neuromuscular control of the knee joint [45]. Also, joint proprioception is positively influenced by long-term training in athletes [46]. In this context, a lack of effective dynamic control results in a progressive decline in knee joint stability during physical activity, ultimately leading to harmful situations that may cause injury [45].

Our study exclusively included female dancers, which may enhance the applicability of the results given the higher injury rate observed among female ballet dancers. A systematic review found that, in professional dancers, the incidence of injury was 1.06 injuries per 1000 dance hours for males and 1.46 injuries per 1000 dance hours for females. Additionally, 64% of female injuries were attributed to overuse, compared with 50% in males, indicating a higher proportion of overuse injuries in female professional ballet dancers [13]. Another study reported that the incidence of injury among Greek professional dancers was 1.10 injuries per 1000 dance hours in males and 1.55 injuries per 1000 dance hours in females, suggesting a higher injury rate in female dancers [47].

Epidemiological evidence links medial meniscus lesions in athletes frequently to acute or chronic anterior cruciate ligament (ACL) tears, with occurrences reported at 82% and 96%, respectively [48]. In a study by Park et al. [49], the risk of ACL injury was investigated in athletes engaged in pivoting activities, measuring knee joint laxity and rotational range in extreme internal and external tibial rotation. They noted significant differences in the maximum passive external tibial rotation between males and females, with excessive tibial rotation being identified as a major risk factor for proprioception loss and injuries like meniscus tears [49,50,51]. Supporting these findings, LaPrade and Burnett [52] characterized knee injuries as resulting from sudden, excessive twisting loads applied to a flexed, planted knee, leading to lateral femoral condyle impingement. Their research, along with cadaveric studies, emphasizes the critical role of knee biomechanics in meniscal displacement and deformation, which is vital for load distribution, shock absorption, and joint congruity during movement [52,53]. Additionally, Furumatsu et al. [54] found that high knee flexion exercises, particularly those involving femoral internal rotation or tibial external rotation, increase stress on the medial meniscus, particularly on its posterior root, heightening the risk of injury-related stress and degeneration [54,55].

Our study has several limitations. The measuring approach utilized in the current research may be considered as inadequate due to certain technological constraints. During the Grand Plié, skin movement might have influenced the markers’ accuracy, thereby introducing an error in measuring the tibial rotation angles, which is the most difficult movement to accurately define [56,57]. The discrepancy in marker readings can be attributed to the relative movement between the skin and the precise location of the underlying anatomical structures. Thus, as the dancer transitioned into the Grand Plié position, the knee’s rotation axis underwent three-dimensional displacement in space. While the point cluster technique has been refined and validated for minimizing artifacts resulting from skin movement, there is still a potential for an overestimation of the rotational range of motion. Regarding our sample, two limitations are that all participants were recruited from the same group (the Greek National Opera Ballet) and there was a broad age range (15.99–34.21 years). These limitations may impact the generalizability of our findings. Also, this study acknowledges the potential influence of cumulative training load, such as soft tissue overuse, particularly due to the broad age range of participants (15.99–34.21 years), which might have affected the results. Moreover, while our study recruited a sample of 20 ballet female dancers, we acknowledge the limitation of not conducting an a priori sample size estimation. This decision was primarily due to the challenges in recruiting a larger group of elite dancers. However, it is noteworthy that our participants were elite performers and members of the Greek National Opera Ballet.

## 5. Conclusions

Our study investigated the knee rotation during the Grand Plié in professional classical ballet dancers, with the aim of understanding its alignment with the knee mechanics observed in similar movements like squats. The findings indicate that internal rotation during the Grand Plié often exceeds typical joint ranges, suggesting a potential risk for soft tissue injuries such as meniscal tears. The current study provides an opportunity for future research and kinematic analyses, focusing on the movement of the Grand Plié and comparable movements. The potential to improve technique and reduce injury risk based on these biomechanical insights can be a valuable focus in subsequent studies. Additional investigations are required regarding kinematic assessments in dance, conducted beyond the limits of controlled laboratory settings, employing inertial measurement units (IMUs) [34]. These data have the potential to yield valuable information about the kinematics of the knee concerning meniscus damage.

## Figures and Tables

**Figure 1 sports-12-00054-f001:**
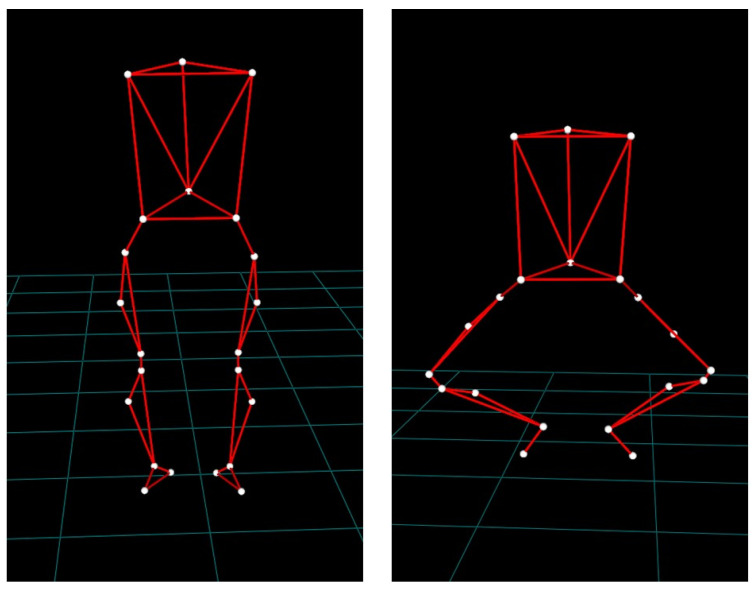
Position at standing and finishing position of Grand Plié.

**Figure 2 sports-12-00054-f002:**
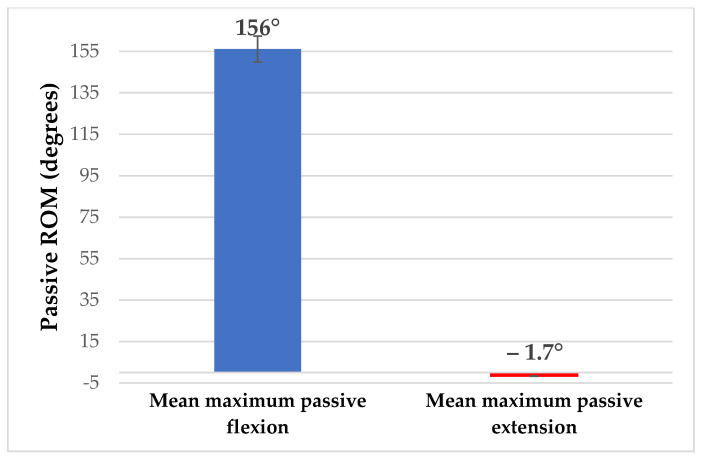
Average passive ROM of the knee flexion and extension.

**Figure 3 sports-12-00054-f003:**
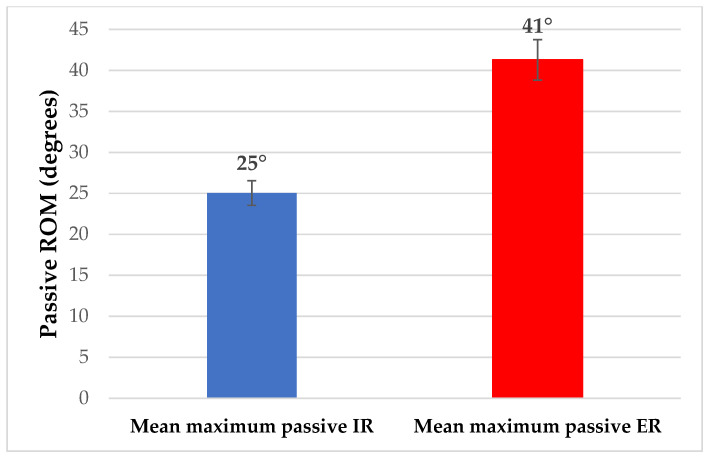
Average passive ROM of the knee internal (IR) and external rotation (ER).

**Figure 4 sports-12-00054-f004:**
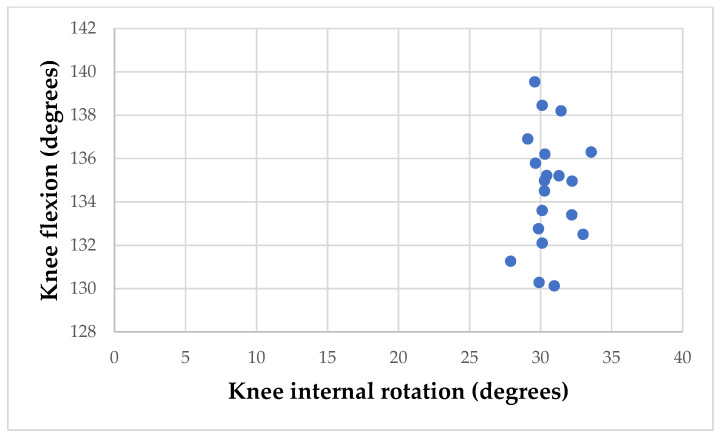
Internal knee rotation and concurrent knee flexion at lowest position of Grand Plié.

**Table 1 sports-12-00054-t001:** Sample demographics (N = 20).

	Mean ± SD	Range (Min–Max)
Age, years	23.58 ± 5.31	15.99–34.21
Height, cm	163.53 ± 5.89	148.00–172.50
Weight, kg	53.95 ± 5.97	41.00–62.50
BMI, kg/m^2^	20.13 ± 1.47	17.19–22.96

SD = standard deviation.

## Data Availability

The data that support the findings of this study are available from the corresponding author, A.F., upon reasonable request. Data is available on request due to privacy and ethical issues.

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
