# Peer review of "Excessive Knee Internal Rotation during Grand Plié in Classical Ballet Female Dancers"

_sports, 2024, doi:10.3390/sports12020054_

Round 1
Reviewer 1 Report
Comments and Suggestions for Authors
This is an interesting study with practical applications. It is well written, but some points need clarification.
The study group should be better presented - how many years they practice the ballet? The age range of the subjects is 15.99-34.21 years, therefore they may differ significantly according to soft tissues overuse, what may influence the results. This problem should be also addressed in the study limitations and in the discussion.
The reliability of all measurements are lacking in the methods - please add them and cite appropriate references.
The Figure 6 should be numbered as 4.
The study limitations section is lacking.
Author Response
Dear Reviewer 1,
We wish you a Happy New Year!
Thanks for the recommendations. Please find attached our responses.

Reviewer 2 Report
Comments and Suggestions for Authors
Thank you for the opportunity to review. Unfortunately I have some serious reservations. As this is the first round of reviews, I do not understand the mode of tracking changes in the document.
1. Why did the authors choose to study only women? (l 103)
2. Add sample size calculations ? (l103)
3. Justify the age range of the study subjects.
4. I do not see clear inclusion and exclusion criteria.
5. ‘’ All statistical analysis performed using a spreadsheet software (Excel, Microsoft Co., USA).’’ – in my opinion, you cannot say that a statistic was carried out in Excel. Let's go back to the research hypothesis: ‘’ The main hypothesis of this study is that knee rotation during a Grand Plié will be comparable to that observed in previous research on squat movements.’’ – specific statistical tests should be added in the studies. There is a lack of information according to this control group.
6. According to figures 2-3 a comparison was carried out. Why was no statistical comparison made? What was the p-value and the effect size.
7. Refrains from a thorough evaluation of the results, discussion and conclusions until the methodology is improved.
Author Response
Dear Reviewer 2,
Thanks for your suggestions and comments. Please find attached our reponses to your comments.
We wish you a Happy New Year!

Reviewer 3 Report
Comments and Suggestions for Authors
Introduction
The authors relate knee joint injuries to athletes. Nevertheless, dancers are not considered athletes. They should rewrite this statement in order to include dancers as a population at risk.
There is a need to define ballet. I mean, what does ballet involve in relation to biomechanichs. How many hours spent professional dancers training? What is the age interval when it comes to professional ballet dancers? In addition, there is a need to justify ballet as a sport modality. It is accepted by the World Dance Federation?
Is it the Grand Plié a transition movement (a standard movement that is usually performed between other movements)?
Is there any importance of sex, or studies that have identified sex as an injury factor risk in ballet?
Methods
Is there any specific reason why only females are included? This should also be reflected in the title.
If possible, it would be useful to know the existence of previous injuries in the sample, as well as the years practicing ballet.
How the participants were recruited? Do they belong to the same ballet group (it seems so as volume of training is identical). Please, determine hours of training.
I am a little bit confused with range values regarding age. The younger participant was 15.99 years old? The older one was 34.21 years? From my understanding age in years is represented as an entire (whole) number.
RESULTS
The manuscript aims at identifying excessive internal rotation. After reading the results, it is not clear if excessive internal rotation is present. How many dancers showed this feature? Were age, years of training/dancing involved in the presence/absence of excessive internal rotation?
Where is the cut-off point for identifying excessive internal/external rotation?
Judging from the David protocol and Figure 1. Both knees were measured. Is there any chance to show internal and external rotation angles for both knees separately? Or both knees reached extremely similar angles?
DISCUSION
Since the sample included only female dancers, the importance of sex as knee injury risk factor should be discussed.
Also, if the participants were recruited from the same group of dancers, this is a limitation to consider.
CONCLUSION
I find this section too long. Please, respond to the hypothesis stated in the introduction and specify whether excessive external/internal rotation exists.
It is stated that the objective is to enhance technique performance of the Grand Pilé. This should be commented in the discussion section. How technique can be improved to reduce injury risk. Should the Grand Pilé be performed differently?
Comments on the Quality of English LanguageNo comments
Author Response
Dear Reviewer 3,
Thanks for your thorough reviewing. Please find attached our point-by-point responses to your comments.
We wish you a Happy New Year!

Round 2
Reviewer 2 Report
Comments and Suggestions for Authors
Thank you for your reply. The authors have responded correctly to my comments number 1, 3, 4, 6.
However, comment 2 and 5 unfortunately I do not accept the answer.
‘’This study aims to investigate knee rotation throughout flexion in the Grand Plié movement in professional classical ballet dancers. ‘’
A small group not supported by statistical calculations does not allow the results to be confirmed. It could just be a coincidence. The lack of a statistical comparison with an additional group of, for example, non-exercisers does not tell us anything about the mobility observed by the authors. In conclusion, the results can only be related to the particular group studied by the authors, n=20. We do not know whether this is due to training or to chance. I stand by my previous decision.
Yours sincerely
Author Response
Dear Reviewer 2,
Thank you once again for your invaluable feedback.
In response to your concerns regarding our sample size, we utilized specific input parameters in our G*power analysis, yielding a post-hoc power of approximately 0.95. These parameters were as follows:
- Tail(s): two
- Effect size d: 0.8571429 (based on constant H0: 25.0, H1: 30.28, SD σ: 6.16 of our study)
- α err prob: 0.05
- Total sample size: 20
We acknowledge the challenges in assembling large sample sizes, particularly when dealing with high-level athletes or populations meeting very specific criteria, such as professional female ballet dancers. If someone makes a search on the Greek National Opera Ballet's website, it reveals that there are currently 21 professional female ballet dancers, closely aligning with our sample size. As we addressed in our earlier response, while post-hoc analysis is generally not recommended, we chose to trasnparently highlight the absence of a priori sample size calculation in our study's limitations, despite the relatively high post-hoc power.
Regarding the results and their potential attribution to coincidence, our scatter plot data demonstrate a consistent pattern in knee rotation among all participants, suggesting a significant trend beyond mere chance. Moreover, although to not have a control group to compare with is a methodological issue, in the context of our study, which is centered on a highly specialized and skill-specific movement like the Grand Plié in classical ballet, the selection of an appropriate control group poses a unique challenge. The specificity and complexity of this movement are such that it would not be meaningful or relevant to compare ballet dancers with individuals who do not have the same level of training or expertise in ballet. It is for this reason that our study focused exclusively on professional ballet dancers. The essence of our research was to provide a biomechanical analysis of the Grand Plié, a movement intrinsic to classical ballet, and to highlight the significant rotational demands it places on the knees. This is a distinctive aspect of ballet that differs considerably from more generic movements like squats, which are common in various forms of physical activity.
We appreciate your critical assessment and hope that our clarification addresses your concerns more thoroughly.
Reviewer 3 Report
Comments and Suggestions for Authors
I thank the authors for the provided responses to my suggestions.
Author Response
Dear Reviewer 3,
Thank you too for your valuable suggestions.